# Growth performance, physiological parameters, and transcript levels of lipid metabolism-related genes in hybrid yellow catfish (*Tachysurus fulvidraco* ♀ × *Pseudobagrus vachellii* ♂) fed diets containing Siberian ginseng

**Ming Xiao Li** [1], **Jun Qiang**[2], **Jing Wen Bao**[2], **Yi Fan Tao**[2], **Hao Jun Zhu**[2], **Pao Xu**[1,2]*

1 Wuxi Fisheries College, Nanjing Agricultural University, Jiangsu, Wuxi, China, 2 Key Laboratory of Freshwater Fishes and Germplasm Resources Utilization, Ministry of Agriculture, Freshwater Fisheries Research Center, Chinese Academy of Fishery Sciences, Jiangsu, Wuxi, China

* Xup@ffrc.cn

**Data Availability Statement:** Raw sequence data have been submitted to the Gene Expression

## Abstract

In high-density aquaculture, fish health can suffer because of excessive feeding, which causes fatty liver disease. Siberian ginseng (*Acanthopanax senticosus*) has been used as a feed additive to promote animal growth, immunity, and lipid metabolism. In this study, we explored the effects of *A. senticosus* on the physiology of hybrid yellow catfish (*Tachysurus fulvidraco* ♀ × *Pseudobagrus vachellii* ♂). A control group and five groups fed diets containing *A. senticosus* (0.5, 1, 2, 4, and 8 g *A. senticosus*/kg feed) were established and maintained for 8 weeks. Dietary supplementation with *A. senticosus* at 4 g/kg promoted growth of the hybrid yellow catfish. Serum total cholesterol (TC) and triacylglycerol (TG) levels at 2 g/kg *A. senticosus* (TC: 1.31 mmol/L; TG: 1.08 mmol/L) were significantly lower than in the control group (TC: 1.51 mmol/L; TG: 1.41 mmol/L), and 4 g/kg *A. senticosus* (17.20 μmol/g tissue) reduced the liver TG level compared with the control group (21.36 μmol/g tissue) ($P$<0.05). Comparative transcriptomic analysis of liver tissue between the control group and the group showing optimum growth (4 g/kg *A. senticosus*) revealed 820 differentially expressed genes and 44 significantly enriched pathways, especially lipid metabolism pathways such as unsaturated fatty acid and fatty acid metabolism. The transcript levels of five lipid metabolism-related genes were determined by quantitative real-time PCR. The results showed that 2–4 g/kg *A. senticosus* supplementation reduced the *FADS2*, *ELOVL2*, *CYP24a*, and *PLPP3* transcript levels and 4 g/kg *A. senticosus* increased the *DIO2* transcript level ($P$<0.05), leading to altered synthesis of TG and thyroxine and reduced fat deposition in the liver. Our results show that dietary *A. senticosus* affects the regulation of fat metabolism and promotes the growth of hybrid yellow catfish. *A. senticosus* is a healthy feed additive, and the appropriate dietary supplementation rate is 2–4 g/kg.

Omnibus at the NCBI under the GEO series accession number GSE154894 (https://www.ncbi.nlm.nih.gov/geo/query/acc.cgi?acc=GSE154894).

**Funding:** This work was supported financially by the Natural Science Foundation of Jiangsu Province, China [Grant no. BK20181137], Project of Six Talent Peaks in Jiangsu Province (NY-133) and the Chinese Academy of Fishery Sciences (Project 2020TD37). Funding was awarded to JQ. The funders had no role in study design, data collection and analysis, decision to publish, or preparation of the manuscript.

**Competing interests:** The authors have declared that no competing interests exist.

**Abbreviations:** *A. senticosus*, *Acanthopanax senticosus*; *CYP24a*, 1,25-dihydroxyvitamin D(3) 24-hydroxylase; *DIO2*, iodothyronine deiodinase 2; *ELOVL2*, ELOVL fatty acid elongase 2; *FADS2*, fatty acid desaturase 2; FCR, feed conversion ratio; GIFT, genetically improved farmed tilapia; HL, hepatic lipase; HSI, hepatosomatic index; LPL, lipoprotein lipase; PER, protein efficiency ratio; *PLPP3*, phospholipid phosphatase 3; PUFAs, polyunsaturated fatty acids; SGR, specific growth rate; T3, triiodothyronine; T4, thyroxine; TC, total cholesterol; TG, triglyceride; TH, thyroid hormone; TP, total protein; VSI, viscerosomatic index; WGR, weight gain rate.

## Introduction

With the development of intensive aquaculture, many factors such as environmental pollution [1], excessive energy intake [2], and a lack of nutrients [3] can lead to fatty liver disease in farmed fish. Fatty liver disease has been shown to damage the structure of liver cells and affect liver metabolism, leading to slow growth and weakened immunity of farmed fish [4–6]. This can cause severe economic losses to the aquaculture industry. Therefore, research on the prevention or treatment of fatty liver in fish is necessary.

Recent studies have suggested that Chinese herbal medicines, which have both nutritional and pharmacological properties, have considerable potential as additives in aquafeeds [7, 8]. Several studies have shown that Chinese herbal medicines can reduce lipid levels in animals and protect against fatty liver disease [9, 10]. Siberian ginseng (*Acanthopanax senticosus*) is a traditional Chinese medicinal plant. Its roots and rhizomes or stems are harvested in the spring and fall, and are dried for use as a medicine [11]. It is rich in nutrients such as trace elements and amino acids, which are required by animals, as well as a variety of active ingredients such as saponins, polysaccharides, and flavonoids [12, 13]. *A. senticosus* is considered to be a high-quality medicinal material that can positively affect animal growth [14], enhance immunity [15], improve antioxidant capacity [16], treat inflammation [17, 18], and regulate lipid metabolism [11, 19]. The physiological role of *A. senticosus* in lipid metabolism has been proven by a large number of experimental studies. For example, cortisol extracts from the stems of *A. senticosus* were shown to reduce lipid synthesis in the liver and reduce the insulin concentration to ameliorate liver steatosis [19]. A metabolomics analysis of mice with Parkinson's disease showed that an extract of *A. senticosus* affected the regulation of long-chain saturated fatty acid mitochondrial β oxidation, fatty acid metabolism, and other metabolic pathways [20]. Our previous research [11] showed that genetically improved farmed tilapia (GIFT, *Oreochromis niloticus*) fed with diets containing *A. senticosus* showed reduced liver fat deposition and lower fat contents, compared with those fed a control diet. Yellow catfish has delicious meat and high nutritional value, and is an important freshwater cultured species in China. Its production was 536,964 tons in 2019, a 5.37% increase over its production in 2018 [21]. The large-scale farming of yellow catfish means the fish is susceptible to high lipid deposition in the liver during breeding, which can affect its immunity and the product quality. This problem poses risks to the yellow catfish breeding industry [22, 23]. However, few studies have explored the effect of *A. senticosus* on regulating lipid metabolism in yellow catfish.

To answer the above question, we conducted experiments to explore the mechanism by which dietary *A. senticosus* supplementation affects lipid metabolism in the liver of hybrid yellow catfish. Transcriptome analyses have been widely used in studies on the immunity, nutrition, and metabolism of aquatic organisms [24, 25]. For example, recent studies have used transcriptome analyses to reveal a series of molecular changes that regulate lipid metabolism in tilapia (*O. niloticus*) [26], salmon louse (*Caligus rogercresseyi*) [27], Atlantic salmon (*Salmo salar*) [28], zebrafish (*Brachydanio rerio*) [29], and the larvae of Atlantic cod (*Gadus morhua*) [30]. In this study, to determine how the Chinese medicine *A. senticosus* affects lipid metabolism in hybrid yellow catfish (*Tachysurus fulvidraco* ♀ × *Pseudobagrus vachellii* ♂), we used the Illumina HiSeq 4000 sequencing platform to detect differences in the liver transcriptomes between the control group and the group showing optimal growth. We detected many differentially expressed genes related to lipid metabolism, and selected five for confirmation of their transcript levels by qRT-PCR. Hepatic steatosis is strongly associated with abnormal lipid metabolism, which is reflected mainly in the imbalance between hepatic lipid inputs and outputs, and blood is an important transport system in the regulation of lipid metabolism [31, 32]. Triglyceride (TG) and total cholesterol (TC) are important components of blood lipids,

and they are mainly synthesized in the liver. Therefore, changes in TG and TC content can be used to assess the fat metabolism of fish. Triiodothyronine (T3) and thyroxine (T4) are thyroid hormones (THs) that are involved in maintaining the normal metabolism of blood lipids. The main physiological functions of lipoprotein lipase (LPL) are to convert TG into very low-density lipoprotein and to promote the transfer of TC and phospholipids between lipoproteins [33]. Hepatic lipase (HL) is synthesized primarily in liver and is involved in chylomicron-remnant and high-density lipoprotein metabolism [34]. Increased LPL and HL activities can affect serum TC and TG levels. Therefore, by measuring TH content and catabolic enzyme activity, changes in the lipid metabolism of yellow catfish can be assessed.

In this study, we screened the differentially expressed genes and enriched pathways related to lipid metabolism in the liver of hybrid yellow catfish by transcriptome sequencing, and measured the changes in serum and liver lipids, TH content, and the activities of catabolic enzymes at different *A. senticosus* supplementation levels. The aim was to explore the mechanism by which *A. senticosus* regulates fat metabolism in the hybrid yellow catfish at the physiological and molecular levels, and to provide a scientific basis for the use of *A. senticosus* as an aquatic feed additive.

## Materials and methods

### Ethics statement

The study protocols were approved by the Ethics Committee at the Freshwater Fisheries Research Centre of the Chinese Academy of Fishery Sciences (FFRC, Wuxi, China). All experimental procedures were performed according to the Guide for the Care and Use of Laboratory Animals in China.

### Experimental diets

In our previous research [11], we found that 1–2 g/kg dietary levels of *A. senticosus* promoted growth, enhanced immunity, and reduced lipid accumulation in GIFT. Therefore, in this experiment, a control group (no dietary *A. senticosus*) and five treatment groups (0.5, 1, 2, 4 and 8 g *A. senticosus*/kg feed) were established (Table 1). We used commercially available *A. senticosus* powder processed by ultra-fine pulverization technology, which was provided by the Beijing Yujing Biotechnology Co. Ltd. (Beijing, China). All ingredients were mixed and then 5% oils and 10% water were added to obtain a dough. The dough was extruded as strands using a laboratory granulator, and the strands were broken into 1.5-mm pellets. The pellets were air dried at 4°C for he and then stored at −20°C until use.

### Experimental facility and fish rearing

The juveniles were chosen from the Yixing base of the FFRC. Juveniles were acclimated in an aerated flow-through system and fed with commercial feed for 1 week. The experiment was carried out in a circulating water system. A total of 450 healthy hybrid yellow catfish juveniles with mean weight of 13.41±0.04 g were selected and randomly divided into six groups, each with three replicate tanks. The tanks were filled with 600 L dechlorinated freshwater, and the flow rate was 200 kg/min. The stocking density was 25 fish per tank. The fish were fed to apparent satiation twice a day at 08:00 and 17:00 for 56 days. During the entire experimental period, the water temperature was maintained at 26–28°C and the pH was 7.4. Dissolved oxygen was ≥5 mg/L, and ammonia-N and nitrite concentrations were kept below 0.01 mg/L. One-third of the water was exchanged every 3 days.

**Table 1. Composition and nutrient levels of experimental diets (as-fed basis) %.**

| Items | Add the amount of *A.senticosus* (g/kg) | | | | | |
|---|---|---|---|---|---|---|
| | 0 | 0.5 | 1 | 2 | 4 | 8 |
| Fish meal | 10.00 | 10.00 | 10.00 | 10.00 | 10.00 | 10.00 |
| Wheat middling | 10.60 | 10.60 | 10.60 | 10.60 | 10.60 | 10.60 |
| Corn starch | 16.80 | 16.80 | 16.80 | 16.80 | 16.80 | 16.80 |
| Soybean oil | 5.00 | 5.00 | 5.00 | 5.00 | 5.00 | 5.00 |
| Soybean meal | 16.00 | 16.00 | 16.00 | 16.00 | 16.00 | 16.00 |
| Cottonseed meal | 16.00 | 16.00 | 16.00 | 16.00 | 16.00 | 16.00 |
| Rapeseed meal | 16.00 | 16.00 | 16.00 | 16.00 | 16.00 | 16.00 |
| Vitamin premix | 0.50 | 0.50 | 0.50 | 0.50 | 0.50 | 0.50 |
| Mineral premix | 0.50 | 0.50 | 0.50 | 0.50 | 0.50 | 0.50 |
| Choline chloride | 0.50 | 0.50 | 0.50 | 0.50 | 0.50 | 0.50 |
| Sodium vitamin C phosphate | 0.20 | 0.20 | 0.20 | 0.20 | 0.20 | 0.20 |
| $Ca(H_2PO_4)^2$ | 1.50 | 1.50 | 1.50 | 1.50 | 1.50 | 1.50 |
| Microcrystalline cellulose | 6.40 | 6.35 | 6.30 | 6.20 | 6.00 | 5.60 |
| *A.senticosus* | 0.00 | 0.05 | 0.10 | 0.20 | 0.40 | 0.80 |
| Total | 100.00 | 100.00 | 100.00 | 100.00 | 100.00 | 100.00 |
| *Proximate composition* (%, DM) | | | | | | |
| Ash | 8.27 | 8.38 | 8.29 | 8.39 | 8.42 | 8.53 |
| Dry matter | 92.61 | 92.60 | 92.94 | 92.88 | 92.83 | 93.02 |
| Crude protein | 28.20 | 28.44 | 28.53 | 28.61 | 28.57 | 28.77 |
| Crude lipid | 6.66 | 6.41 | 6.61 | 6.72 | 6.49 | 6.33 |

## Sample collection

After the 56-day feeding experiment, the fish were starved for 24 h to reduce the effects of feeding on physiological and biochemical indicators. The fish were treated with 100 mg/L MS-222 (Argent Chemical Laboratories, Redmond, WA, USA) for rapid deep anesthesia, and the total weight of all fish in each tank was recorded. Four fish were randomly caught from each tank, and blood samples were taken from the tail vein and immediately centrifuged to obtain serum as described by Ma [35]. The serum was stored in labeled tubes at −80˚C for further biochemical analyses. Four fish per tank were weighed and dissected, and then their liver and viscera tissues were weighed. The liver and intestinal tissues were collected from 12 fish in each group, frozen in liquid nitrogen, and stored at −80˚C. The intestine samples were used to determine the activities of digestive enzymes. The liver samples were separated into two portions, one of which was used to determine physiological indexes, and the other for RNA extraction. According to growth and biochemical indicator data, further analyses were conducted to compare the livers of the group showing optimal growth (T_Liver) with those of the control group (C_Liver). Three fish from each group were randomly selected from each tank, and the liver tissues were quickly dissected, frozen in liquid nitrogen, and stored at −80˚C until transcriptome sequencing.

## Fish growth performance

Fish growth performance was determined by calculating the weight gain rate (WGR), specific growth rate (SGR), viscerosomatic index (VSI), hepatosomatic index (HSI), and protein efficiency ratio (PER). The feed conversion ratio (FCR) was calculated to express feed utilization. During the experiment, the daily feed consumption and the number of fish deaths were recorded. These parameters were calculated according to Li [11].

## Blood biochemical analysis

We measured serum total protein (TP), triglyceride (TG), and total cholesterol (TC) contents using a fully automatic biochemical analyzer (bs-400, MINDRAY, Shenzhen, China). Reagents and test kits were purchased from MINDRAY. The contents of triiodothyronine (T3) and thyroxine (T4) were determined by radioimmunoassay (RIA) as described by Nayak and Singh [36].

## Hepatic lipid index assays

Liver samples (about 1.0 g) were homogenized in ice-cold phosphate-buffered saline (PBS, 50 mmol/ L, pH 7.3) and then centrifuged for 10 min at 3500 g at 4°C [37]. Liver TG, TC, glycogen contents and LPL, HL activities were measured using ELISA kits. All kits were provided by the Shanghai Lengton Bioscience Co., Ltd. (Shanghai, China).

## Enzyme activity analyses

Intestinal samples were homogenized in ice-cold PBS (50 mmol/ L, pH 7.3) and then centrifuged for 10 min at 3500 g at 4°C [37]. The supernatant was used for analyses of amylase, trypsin, and lipase activity as described by Qiang [37].

## Analysis of transcriptome libraries

**RNA extraction and Illumina library preparation.** Total RNA was extracted from livers of fish from the two groups using Trizol reagent (Invitrogen, Carlsbad, CA, USA) according to the manufacturer's instructions. The extracted RNA was detected using a Bioanalyzer 2100 and RNA 6000 Nano LabChip Kit (Agilent, CA, USA). Each group of RNA samples with good integrity and purity was mixed, then approximately 10 μg total RNA was taken from each mixed group to prepare an Illumina library. Six libraries were constructed: T_Liver_1, T_Liver_2, T_Liver_3, C_Liver_1, C_Liver_2, and C_Liver_3. The RNA was reverse-transcribed according to the operating procedures of the mRNA-Seq sample preparation kit (Illumina, San Diego, CA, USA) to create the final complementary DNA (cDNA) library. Each library was sequenced on the Illumina HiSeq 4000 platform. The average insert size was $300 \pm 50$ bp.

**Data filtering, read mapping, and detection of differentially expressed genes.** Invalid reads (including joint, duplicate, and low-sampling reads) were removed from the sequence data using the processing steps described by Qiang [38]. The remaining clean reads were used for subsequent analyses. Reads from the T_Liver and C_Liver libraries were aligned to the *Tachysurus fulvidraco* reference genome (https://www.ncbi.nlm.nih.gov/genome/?term= yellow+catfish) using the HISAT package [39]. This software package maps reads to a reference genome to build a database, and compares previously unmapped reads with a database of putative junctions to confirm them.

In the clean RNA-seq data, gene transcript levels were estimated on the basis of fragments per kilobase of transcript per million mapped (FPKM) reads values [40]. Differentially expressed (DE) genes between the T_Liver and C_Liver groups were identified using the R package [41] with the following criteria: adjusted $|\log_2 foldchange| \geq 1.0$ and *P*-value $< 0.05$. The DE genes were subjected to KEGG pathway enrichment analysis, and those with $P < 0.05$ were considered to be significantly enriched.

**Verification of selected DE genes by quantitative real-time PCR.** The KEGG enrichment analysis indicated that dietary *A. senticosus* affected signaling pathways related to lipid metabolism. Therefore, we selected five known genes in this pathway for validation by

**Table 2. Sequences of primers used for qRT-PCR.**

| Name | Primer sequence (5'-3') |
|---|---|
| *FADS2* | F: 5'– ATTGGTTCAGCGGCCATCTT–3' |
| | R: 5'– AGACTGGTTGGGGGCAAAAA–3' |
| *ELOVL2* | F: 5'– GAGTGCATCCCCTACCCAAC–3' |
| | R: 5'– GCAGCATGTCAGCCCTATCT–3' |
| *CYP24a* | F: 5'– ACGCACGAGCTAAAGTGAGA–3' |
| | R: 5'– CCGTTCCTACCAGCCGATTT–3' |
| *PLPP3* | F: 5'– CCAGAATCAGCCTGTGGAGTA–3' |
| | R: 5'– AGTGTGTGCAGTCGTAAGGG–3' |
| *DIO2* | F: 5'– GAACTGTTCCCGTTCGAGGT–3' |
| | R: 5'– TACGATGCACACCCTTTCGT–3' |
| *β-actin* | F: 5'– GGATTCGCTGGAGATGATG–3' |
| | R: 5'– TCGTTGTAGAAGGTGTGATG–3' |

quantitative real-time PCR. The five genes were *FADS2* (encoding fatty acid desaturase 2), *ELOVL2*, (encoding ELOVL fatty acid elongase 2), *CYP24a* (encoding 1,25-dihydroxyvitamin D(3) 24-hydroxylase), *PLPP3* (encoding phospholipid phosphatase 3), and *DIO2* (encoding iodothyronine deiodinase 2). All primers used to amplify the DE genes (Table 2) were synthesized by the Suzhou GeneWiz Biotechnology Co. Ltd (Suzhou, China). Total RNA was extracted from liver samples using Trizol reagent ((Invitrogen) and was reverse-transcribed into cDNA using Prime Script™ RT Master Mix (Takara, Dalian, China). The qRT-PCR analyses were performed in accordance with the instruction manual of the SYBR® Premix Ex Taq (Takara) kit using an CFX96™ Real-time PCR System (Bio-Rad, Hercules, CA, USA). The internal reference gene was *β-actin*. Each PCR mixture (25 μL) consisted of 8 μL RNase-free water, 12.5 μL SYBR Premix Ex Taq II (2×), 0.5 μL ROX Dye (50×), 2 μL forward and reverse primers (10 μM), and 2 μL cDNA working solution. The amplification conditions were as follows: 95˚C for 30 s, followed by 40 cycles of 95˚C˚C for 5 s and 60˚C for 30 s [42]. Each reaction was repeated three times. The relative gene transcript levels in the different treatments were calculated using the $2^{-\Delta\Delta Ct}$ method [43].

## Data analysis

The results are reported as mean ± standard error. Shapiro-Wilk's test and Levene's test were used to test for normal data distribution and homogeneity of variance. Significant differences among treatments were determined by one-way ANOVA with *post-hoc* Duncan's multiple range test. The level of significance was $P < 0.05$. Statistical analyses were conducted using SPSS ver. 22.0 (IBM Corp., Armonk, NY, USA).

## Results

### Growth performance, FCR, and survival

The WGR was highest in the group fed with a diet containing 4 g/kg *A. senticosus* and lowest in the control group (0 g *A. senticosus*) (Table 3). The WGR differed significantly between those two groups ($P < 0.05$). The trend in SGR among the different treatments was similar to that of WGR, and both showed the highest values in the group fed with a diet containing 4 g/kg *A. senticosus* (Fig 1). Compared with the control group, the group fed with a diet containing 4 g/kg *A. senticosus* showed a significantly lower HSI level ($P < 0.05$). However, the VSI, FCR and PER did not differ significantly among the treatment groups or between the treatment

**Table 3. Effect of *A.senticosus* supplementation on growth performance of hybrid yellow catfish (means ± *SEM*).**

| *A.senticosus* (g/kg) | 0 | 0.5 | 1 | 2 | 4 | 8 |
|---|---|---|---|---|---|---|
| IBW (g) | 13.44±0.02 | 13.38±0.03 | 13.43±0.02 | 13.43±0.01 | 13.38±0.00 | 13.39±0.02 |
| FBW (g) | 38.66±0.27$^a$ | 39.84±0.92$^{ab}$ | 41.46±0.71$^{bc}$ | 41.56±0.52$^{bc}$ | 42.68±0.89$^c$ | 39.44±0.71$^{ab}$ |
| WGR (%) | 187.65±1.90$^a$ | 197.63±6.67$^{ab}$ | 208.57±4.84$^{bc}$ | 209.49±3.50$^{bc}$ | 218.97±6.64$^c$ | 194.64±4.92$^{ab}$ |
| SGR (%/d) | 1.89±0.01$^a$ | 1.95±0.04$^{ab}$ | 2.01±0.03$^{bc}$ | 2.02±0.02$^{bc}$ | 2.07±0.04$^c$ | 1.93±0.03$^{ab}$ |
| HSI (%) | 1.58±0.06$^c$ | 1.49±0.06$^{bc}$ | 1.4±0.08$^{abc}$ | 1.34±0.07$^{ab}$ | 1.24±0.08$^a$ | 1.33±0.07$^{ab}$ |
| VSI (%) | 14.17±0.42 | 14.3±0.43 | 13.72±0.55 | 14.38±0.52 | 14.43±0.42 | 13.46±0.43 |
| FCR | 2.13±0.03 | 2.08±0.06 | 2.02±0.07 | 2.01±0.06 | 2±0.06 | 2.1±0.04 |
| PER | 1.67±0.05 | 1.69±0.05 | 1.74±0.06 | 1.73±0.06 | 1.75±0.05 | 1.69±0.05 |
| SR (%) | 95.67±1.33 | 97.33±1.33 | 95.67±1.33 | 97.33±2.67 | 98.67±1.33 | 96±2.31 |

Note. Data with different superscript lowercase letters in the same row indicate a significant difference ($P < 0.05$).

groups and the control group ($P > 0.05$). Dietary supplementation with 0–8 g/kg *A. senticosus* did not significantly affect the survival of hybrid yellow catfish.

## Serum biochemical parameters

The TC level was significantly ($P < 0.05$) lower in the group fed with a diet containing 2 g/kg *A. senticosus* than in the control group (Table 4). The serum TG levels decreased significantly ($P < 0.05$) with increasing amounts of *A. senticosus* in the diet, and were significantly lower in the 2 and 4 g/kg *A. senticosus* treatment groups than in the other groups. There was no significant difference in TP content between the control group and the treatment groups ($P > 0.05$).

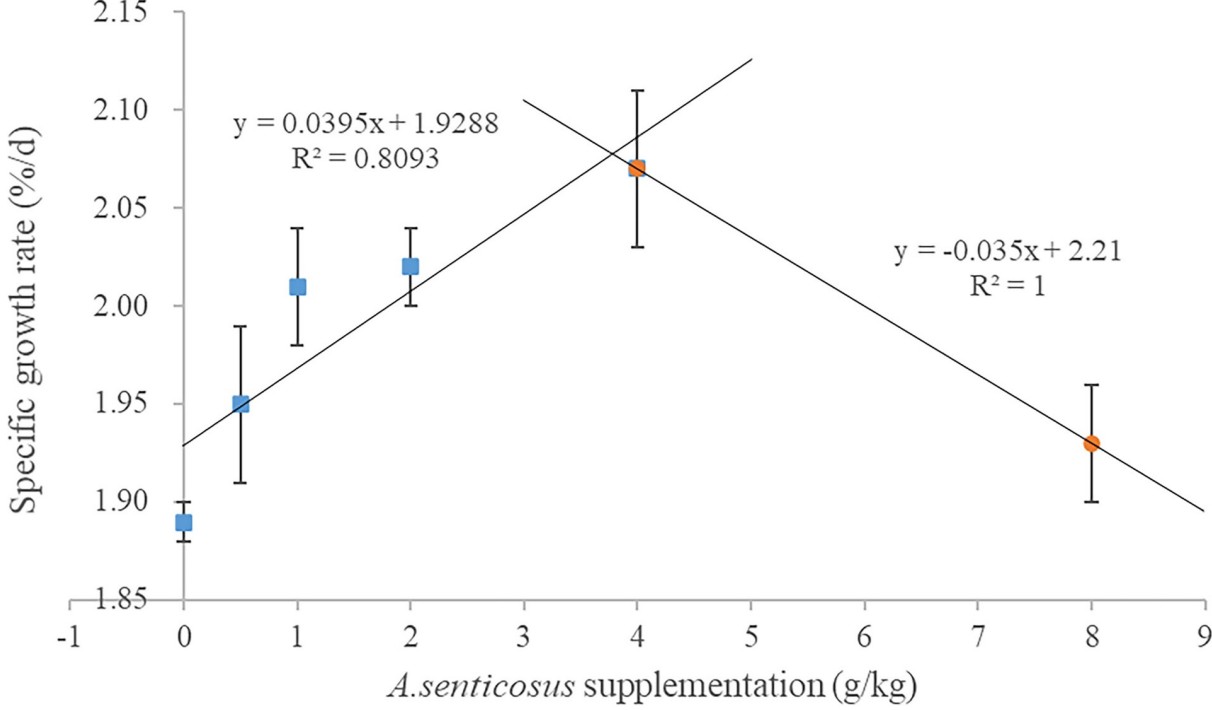

**Fig 1. Relationship between specific growth rate and dietary supplementation with *A. senticosus*.** The relationship was determined using a two-slope broken-line model according to the method of Qiang et al. [38].

**Table 4. Effect of *A.senticosus* supplementation on serum biochemical parameters of hybrid yellow catfish (means ± *SEM*).**

| *A.senticosus* (g/kg) | 0 | 0.5 | 1 | 2 | 4 | 8 |
|---|---|---|---|---|---|---|
| TP (g/L) | 10.08±0.18 | 10.4±0.20 | 10.47±0.21 | 10.55±0.22 | 10.71±0.17 | 10.51±0.25 |
| T3 (mg/mL) | 1.65±0.07[b] | 1.5±0.10[ab] | 1.46±0.09[ab] | 1.29±0.08[a] | 1.25±0.09[a] | 1.47±0.11[ab] |
| T4 (mg/mL) | 59.12±3.32[a] | 67.94±4.24[ab] | 73.42±5.57[ab] | 80.81±4.98[b] | 75.29±4.52[b] | 70.41±5.11[ab] |
| T3/T4 (%) | 2.91± 0.22[b] | 2.36±0.27[ab] | 2.17±0.26[a] | 1.69±0.16[a] | 1.77±0.21[a] | 2.27±0.28[ab] |
| TC (mmol/L) | 1.51±0.05[b] | 1.47±0.05[b] | 1.44±0.05[ab] | 1.31±0.04[a] | 1.38±0.05[ab] | 1.46±0.05[ab] |
| TG (mmol/L) | 1.41±0.07[b] | 1.25±0.07[ab] | 1.25±0.07[ab] | 1.08±0.06[a] | 1.18±0.04[a] | 1.23±0.06[ab] |

Note. Data with different superscript lowercase letters in the same row indicate a significant difference (*P*< 0.05).

Compared with the control group, the groups fed with diets containing 2 and 4 g/kg *A. sentico-sus* showed significantly (*P* < 0.05) lower serum T3 contents and T3/T4 values, and significantly increased serum T4 contents (*P* < 0.05) (Fig 2).

## Hepatic biochemical parameters

The liver TG level differed significantly (*P* < 0.05) between the control group and the group fed with a diet containing 4 g/kg *A. senticosus*, but the TC level did not differ significantly between these two groups (*P* > 0.05) (Fig 3). The glycogen level did not differ significantly

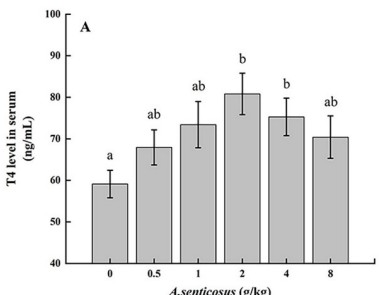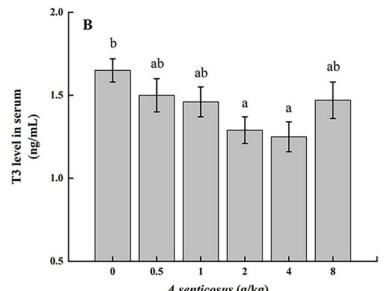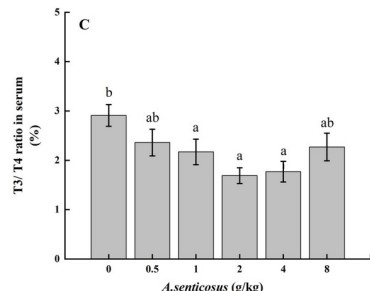

**Fig 2. Serum levels of thyroid hormones in hybrid yellow catfish fed diets with *A. senticosus* supplementation.** (A) Serum thyroxine (T4) and (B) serum triiodothyronine (T3) levels, and (C) T3:T4 ratios with diets with different *A. senticosus* supplementation levels. Different superscript lowercase letters indicate a significant difference (*P* <0.05).

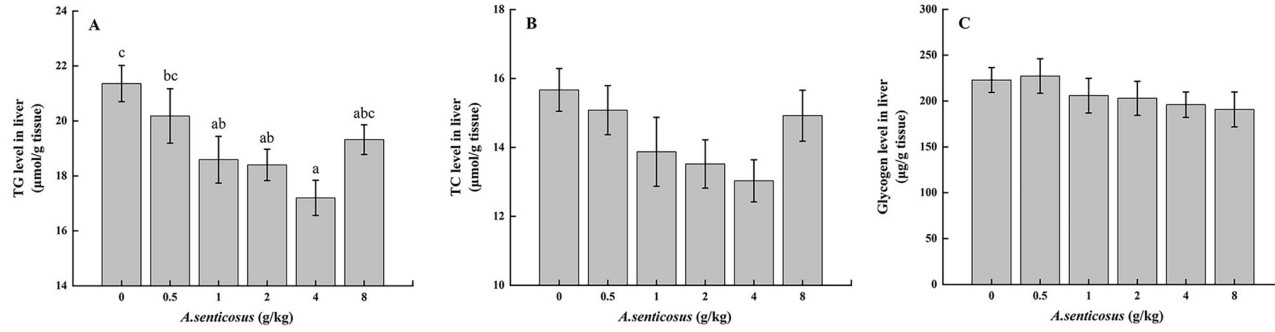

**Fig 3. Liver lipid and glycogen levels in hybrid yellow catfish fed diets with *A. senticosus* supplementation.** (A) Triglyceride (TG), (B) total cholesterol (TC), and (C) glycogen levels with diets with different *A. senticosus* supplementation levels. Different superscript lowercase letters indicate a significant difference (*P* <0.05).

**Table 5. Effect of *A. senticosus* supplementation on catabolic enzymes in the liver of hybrid yellow catfish (means ± *SEM*).**

| *A.senticosus* (g/kg) | 0 | 0.5 | 1 | 2 | 4 | 8 |
|---|---|---|---|---|---|---|
| LPL (ng/mg prot) | 7.39±0.56[a] | 7.77±0.60[a] | 8.8±0.34[ab] | 8.98±0.48[ab] | 9.55±0.53[b] | 8.05±0.50[ab] |
| HL (ng/mg prot) | 31.45±1.02[a] | 32.64±0.81[a] | 33.85±1.16[ab] | 34.23±0.87[ab] | 35.83±0.92[b] | 33.25±0.83[ab] |

**Table 6. Effect of *A.senticosus* supplementation on intestinal digestive enzymes of hybrid yellow catfish (means ± *SEM*).**

| *A.senticosus* (g/kg) | 0 | 0.5 | 1 | 2 | 4 | 8 |
|---|---|---|---|---|---|---|
| Amylase (ng/mg prot) | 96.54±7.38[a] | 103.32±6.17[ab] | 114.93±8.27[ab] | 117.41±8.23[ab] | 126.06±9.47[b] | 153.63±9.48[c] |
| Trypsin (ng/mg prot) | 113.48±9.52[a] | 134.38±6.07[ab] | 142.2±8.81[ab] | 152.39±11.00[b] | 164.26±12.78[b] | 197.73±9.51[c] |
| Lipase ng/mg prot) | 48.61±4.49[a] | 54.58±2.76[ab] | 58.51±4.81[ab] | 61.85±4.42[ab] | 67.33±4.40[b] | 83.48±4.84[c] |

Note. Data with different superscript lowercase letters in the same row indicate a significant difference ($P< 0.05$).

among the treatment groups or between the treatment groups and the control group ($P > 0.05$). The activities of LPL and HL were significantly lower ($P <0.05$) in the group fed a diet containing 4 g/kg *A. senticosus* compared with the control group (Table 5).

## Intestinal digestive enzyme activities

The amylase, trypsin, and lipase activities in the yellow catfish intestine significantly increased with increasing amounts of *A. senticosus* in the diet (Table 6). The amylase, trypsin, and lipase activities were significantly higher in the group fed with a diet containing 8 g/kg *A. senticosus* than in the other five groups ($P < 0.05$).

## Sequence data summary and transcriptome assembly statistics

The Illumina HiSeq 4000 platform was used to sequence the transcriptomes of liver tissues in the T_Liver_1, T_Liver_2, T_Liver_3, C_Liver_1, C_Liver_2, and C_Liver_3 groups. After deleting low-quality sequences, the number of valid reads in each library ranged from 46001180 to 51777748. Across the six groups, the Q20 value was 99.98%, and the GC content was 46% (Table 7). These results confirmed that the sequencing data were of sufficient quality for subsequent splicing and assembly.

**Table 7. Overview of reads for mRNA-seq of hybrid yellow catfish (*Tachysurus fulvidraco* ♀× *Pseudobagrus vachellii* ♂) and quality filtering.**

| Sample | Raw Reads | Base | Valid Read | Base | Valid Ratio (reads) | Q20% | Q30% | GC content % |
|---|---|---|---|---|---|---|---|---|
| C_Liver_1 | 48948446 | 7.34G | 46001180 | 6.90G | 93.98 | 99.98 | 98.53 | 46 |
| C_Liver_2 | 55870092 | 8.38G | 51777748 | 7.77G | 92.68 | 99.98 | 98.66 | 46 |
| C_Liver_3 | 52351492 | 7.85G | 45710806 | 6.86G | 87.32 | 99.98 | 98.72 | 46 |
| T_Liver_1 | 55845642 | 8.38G | 46916382 | 7.04G | 84.01 | 99.98 | 98.61 | 46 |
| T_Liver_2 | 54223560 | 8.13G | 49577940 | 7.44G | 91.43 | 99.98 | 98.59 | 46 |
| T_Liver_3 | 51742012 | 7.76G | 47436396 | 7.12G | 91.68 | 99.98 | 98.50 | 46 |

Note. T_liver: 4 *A.senticosus* / kg; C_liver: 0 *A.senticosus* / kg.

## Differentially expressed genes in different groups

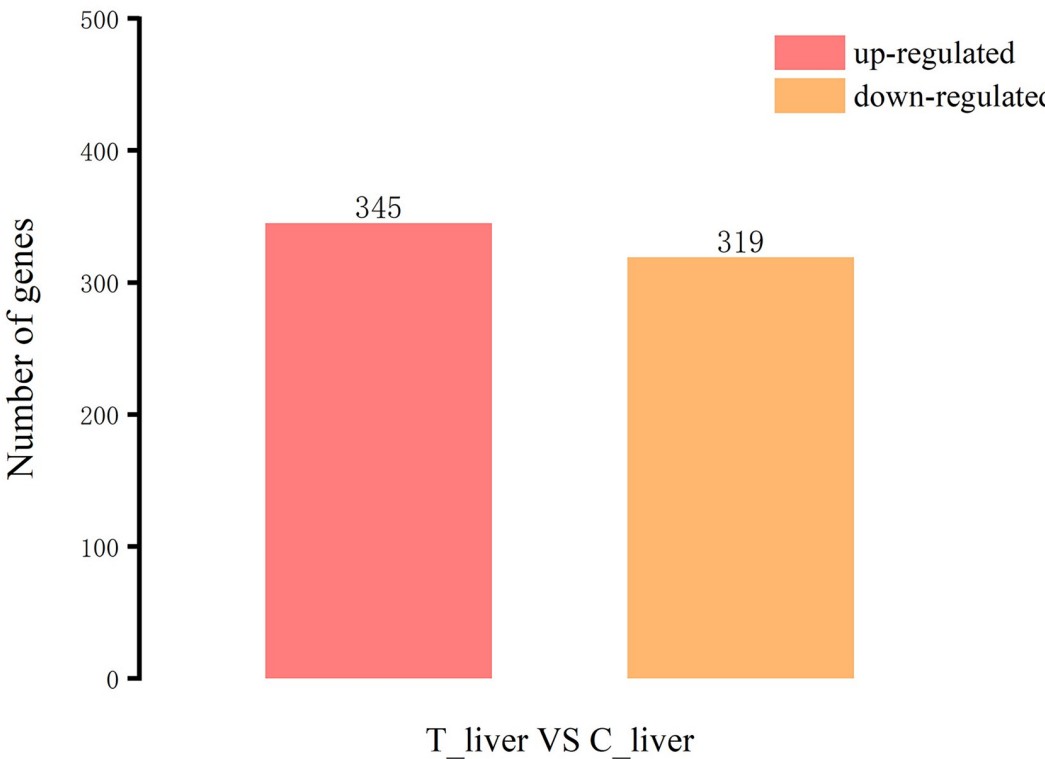

**Fig 4. Expression profiles of genes associated with response to *A. senticosus* supplementation.** The numbers of up- and down-regulated differentially expressed (DE) genes between the group fed the diet with 4 g/kg *A. senticosus* supplementation (T_liver group) and the control group with 0 g/kg supplementation (C_liver group) are shown.

### Screening of differentially expressed genes in response to dietary A. senticosus

We estimated the abundance of genes mapped to the *Tachysurus fulvidraco* reference genome, and identified DE genes based on the following criteria: $|\log_2(\text{foldchange})| \geq 1$ and *P*-value $\leq$ 0.05. The comparison of T_liver and C_liver transcriptomes identified 345 up-regulated genes and 319 down-regulated genes (Fig 4).

### KEGG analysis

The specific signaling pathways enriched with DE genes were identified using tools at the KEGG database. Compared with the C_liver group, T_liver group had multiple pathways enriched with DE genes. The top seven enriched pathways were steroid biosynthesis, biosynthesis of unsaturated fatty acids, alpha-linolenic acid metabolism, thyroid hormone signaling, fatty acid metabolism, propanoate metabolism, and glycerolipid metabolism (Fig 5). Therefore, pathways related to fat metabolism were affected in the yellow catfish fed with diets containing *A. senticosus*.

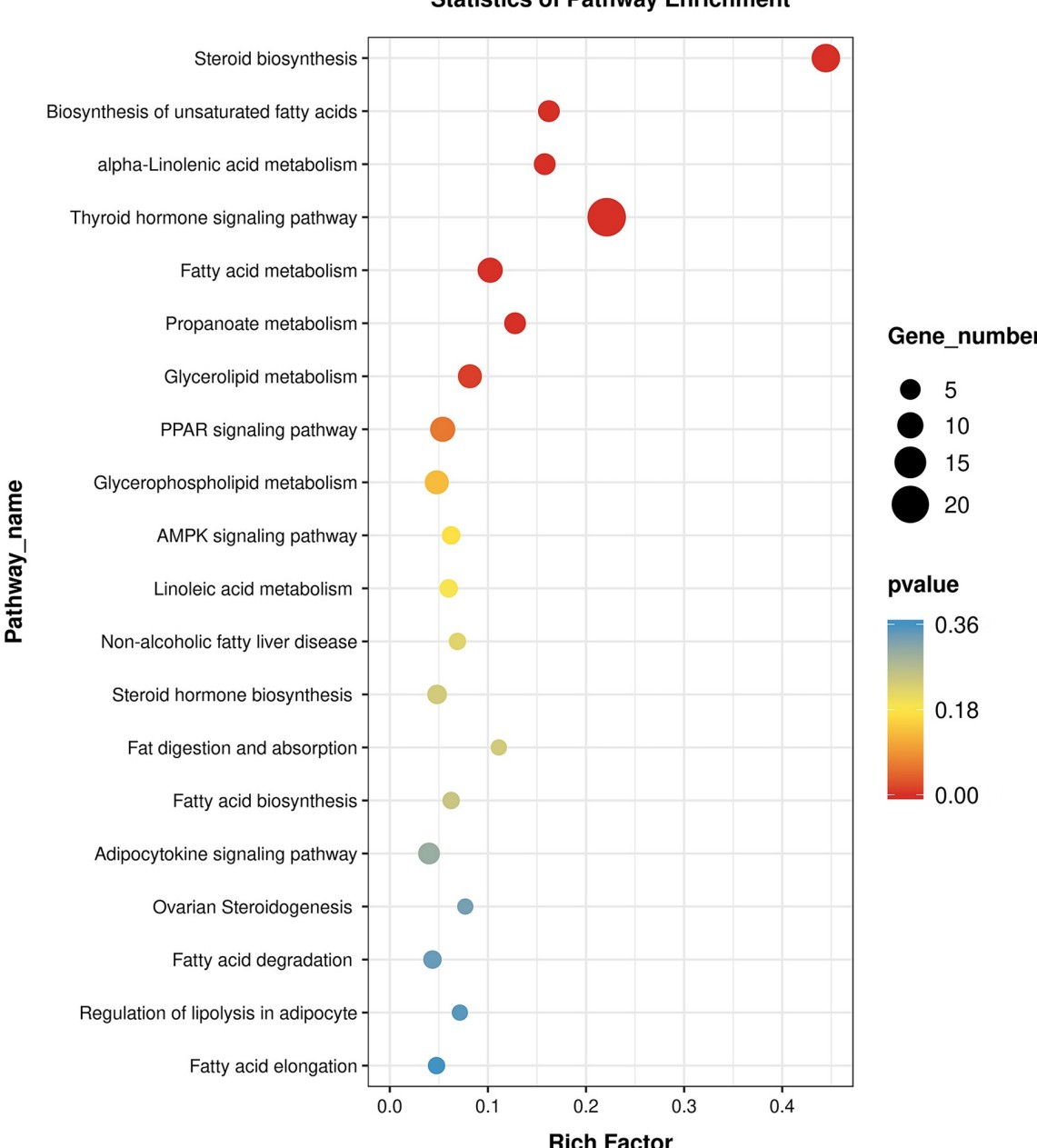

**Fig 5. KEGG analysis of differentially expressed genes (*P* <0.05) associated with response to *A. senticosus* supplementation.** *A. senticosus* supplementation was 4 g/kg in the T_liver group and 0 g/kg in the C_liver group.

## Validation of DE genes

Five DE genes related to lipid metabolism were selected from the enriched pathways for validation by qRT-PCR analyses. Their relative transcript levels were determined in fish fed with diets containing *A. senticosus* at different levels (Fig 6). In the control group and the 4 g/kg treatment group, the transcript levels of genes detected by qRT-PCR were similar to those estimated from the sequencing analysis. *FASD2*, *ELOVL2*, *PLPP3*, and *DIO2* showed decreased transcript levels in fish fed with diets containing *A. senticosus*. Compared with the control

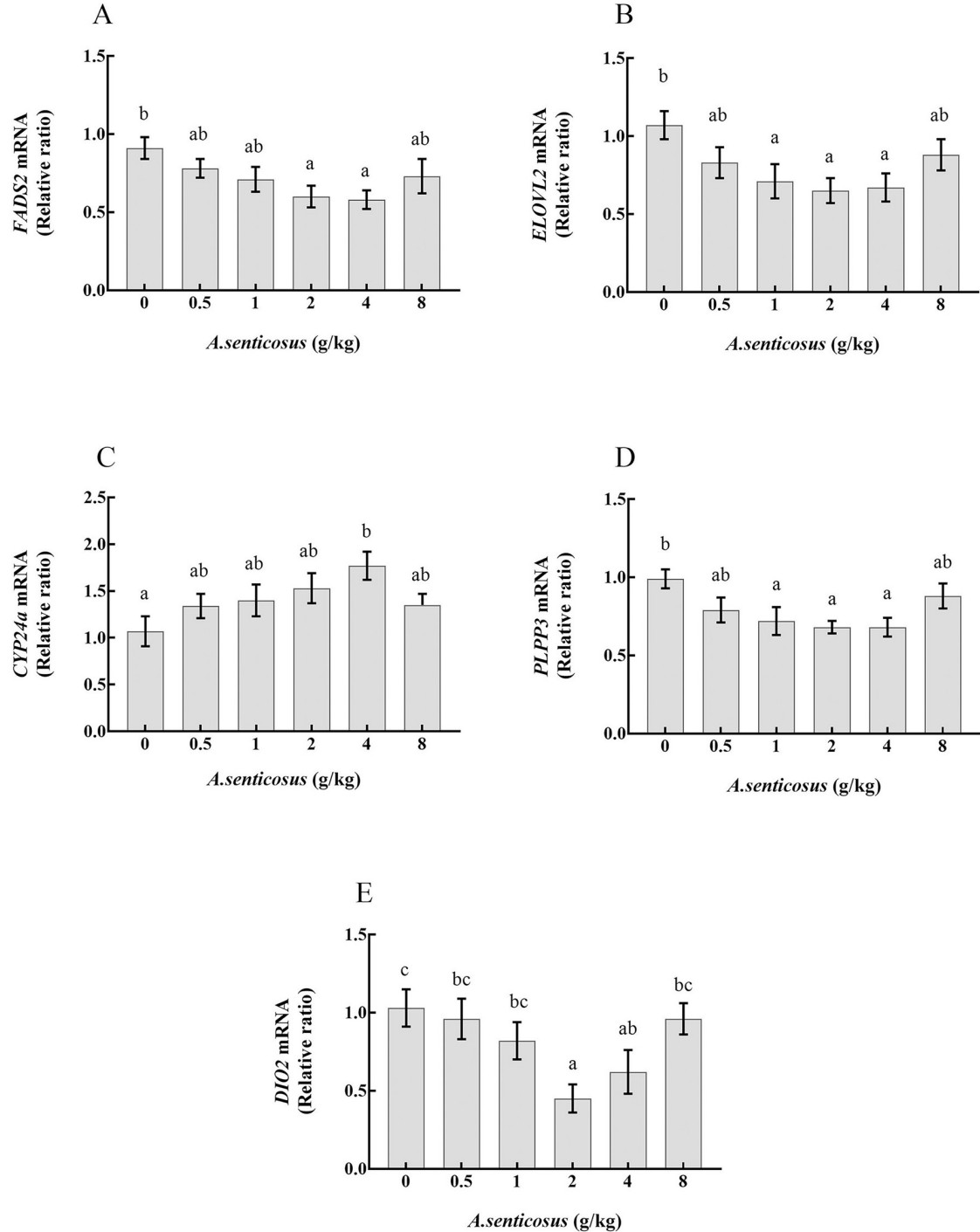

**Fig 6. Expression levels of five differentially expressed genes associated with response to *A. senticosus* supplementation.** The expression levels of (A) *FADS2*; (B) *ELOVL2*; (C) *CYP24a*; (D) *PLPP3*; and (E) *DIO2* in liver of yellow catfish fed diets with different *A. senticosus* supplementation levels are shown. Different superscript lowercase letters indicate a significant difference ($P < 0.05$).

group, the groups fed with diets containing 2 and 4 g/kg *A. senticosus* showed significantly decreased transcript levels of *FASD2* in the liver ($P < 0.05$). Those in groups fed with diets containing 1, 2, and 4 g/kg *A. senticosus* showed the lowest transcript levels of *ELOVL2* and *PLPP3*, significantly lower than in the control group ($P < 0.05$). Compared with the control

group, the group fed with a diet containing 4 g/kg *A. senticosus* showed significantly increased ($P< 0.05$) *CYP24a* transcript levels in the liver. Compared with the control, the groups fed with diets containing 2 and 4 g/kg *A. senticosus* showed significantly decreased ($P< 0.05$) transcript levels of *DIO2* in the liver.

## Discussion

Chinese herbal medicines contain a variety of natural nutrients and biologically active substances, which can promote the metabolism of animals. Various compounds in Chinese herbal medicines can promote the synthesis of proteins and enzymes, accelerate the absorption and utilization of nutrients, and benefit animal growth [44, 45]. Previous studies have shown that *A. senticosus* can promote animal growth. For example, Ruan [46] found that *A. senticosus* significantly promoted the growth of the swamp eel, *Monopterus albus*. An extract of *A. senticosus* was found to significantly improve the growth performance of weaned piglets [47]. Our results show that adding an appropriate amount of *A. senticosus* to the diet can significantly promote the growth of hybrid yellow catfish, and 4 g/kg *A. senticosus* is the optimum level of supplementation. Some studies have suggested that herbal medicines may improve the production performance of livestock via beneficial effects on the activities of intestinal enzymes [48]. *A. senticosus* contains a variety of nutrients and active ingredients, and has been shown to promote the development of animal intestines, promote secretion from digestive glands, and improve feed utilization [49]. In this study, we found that the addition of *A. senticosus* to the diet can significantly improve the digestive function of the intestines of hybrid yellow catfish, leading to better absorption and utilization of nutrients and increased growth. The 8 g/kg *A. senticosus* treatment had the strongest effect intestinal digestive enzyme activity in this study. And the PER did not show a significant difference, but it tended to increase with the *A. senticosus* diet, the highest at 4g/kg. We considered that the digestive enzyme activities and the PER show a positive connection with the appropriate amount of *A. senticosus* diet. However, when excessive *A. senticosus* supplementation, physiological factors such as lipid accumulation may cause this relationship to be insignificant.

As an important metabolic transport system, blood participates in the regulation of lipid metabolism in fish, and blood lipid levels can reflect fat metabolism [50]. In this study, the serum TC and TG contents were significantly lower in the groups fed with diets containing 2–4 g/kg *A. senticosus* than in the other groups and the control. The liver is an important organ for lipid synthesis and storage [51]. Park [19] found that cortisol extracts from the stems of *A. senticosus* reduced lipid synthesis in the liver and reduced the insulin concentration, leading to amelioration of liver steatosis. Sui [52] reported that the addition of *A. senticosus* to aquatic animal feed prevented tissue injury by reducing fat deposition in the body tissue. In this experiment, compared with the fish in the control group, those in the group fed with a diet containing 4 g/kg *A. senticosus* showed significantly lower liver TG levels, significantly reduced serum TG and TC contents, and reduced lipid levels in the body. The HSI level was significantly higher in the control group than in the treatment groups, which reflected fat deposition in the liver and damage to liver cells [53, 54]. These findings are similar to those obtained in our study on GIFT [11]. LPL, a key enzyme in lipoprotein metabolism, can hydrolyze triglycerides circulating in the form of chylomicrons and very low-density lipoproteins [55]. HL is primarily expressed in liver and is involved in chylomicron-remnant and high-density lipoprotein metabolism [56]. In this study, the LPL and HL activities in the group fed with 4 g/kg *A. senticosus* were significantly lower than in the control group, and at this supplementation level, the liver TG and serum TG and TC contents were significantly reduced. Our results suggest

that an appropriate amount of *A. senticosus* increased the activity of TG and TC catabolic enzymes in the liver of hybrid yellow catfish, leading to reduced lipid levels.

In organisms, thyroid hormone (TH) can stimulate lipogenesis in the liver and adipose tissue, and this has major effects on lipid metabolism [57]. In this study, we found that the serum T4 content was significantly increased in the groups fed with diets containing 2 and 4 g/kg *A. senticosus*. An increase in TH can increase fat decomposition and fatty acid oxidation by glucagon, and reduce fat accumulation and the concentration of lipids in the blood. This is consistent with the significantly lower serum and liver TG contents in the groups fed with diets containing 2 and 4 g/kg *A. senticosus* than in the other groups and the control. However, no significant difference in liver glycogen was detected with 0–8 g/kg *A. senticosus* supplementation, possibly because of regulation of the physiological balance of yellow catfish.

To explore the mechanism by which *A. senticosus* affects lipid metabolism in the liver of hybrid yellow catfish, we conducted comparative transcriptome analyses. Then, we validated the results by determining the transcript levels of five DE lipid metabolism genes by qRT-PCR. We selected genes related to fat metabolism and fatty liver disease for these analyses, including *FADS2* encoding Δ-6 desaturase and *ELOVL2* encoding long chain fatty acid elongase, which are important rate-limiting enzymes in the synthesis of polyunsaturated fatty acids [58, 59]. Ma [60] found that interference with *FADS2* expression caused a decrease in intracellular TG content. In another study, overexpression of *ELOVL2* led to increased TG synthesis and accumulation of lipid droplets [61]. In our study, we found that dietary supplementation with *A. senticosus* led to reduced transcript levels of *FADS2* and *ELOVL2* in the liver, which may explain the decreased TG and TC contents. The high transcript levels of *FADS2* and *ELOVL2* in the C_liver may have led to the formation of fat droplets, consistent with the significantly increased HSI, TG, and TC contents in the control group.

Previous studies have shown that, in aquatic animals, the expression of cytochrome P450 family 24 (*CYP24*) can regulate the catabolism of the active form of vitamin D (1, 25-$(OH)_2D_3$) [62], and that 1, 25-$(OH)_2D_3$ can inhibit the differentiation of preadipocytes and affect the transcript levels of genes related to fat metabolism [63]. Zhou and Mai [64] found that a lack of vitamin $D_3$ led to increased fat content in abalone (*Haliotis discus hannai*). In addition, obese patients with nonalcoholic fatty liver disease were found to have low vitamin D levels [65]. These findings and observations suggest that *CYP24a* regulates lipid metabolism by affecting the function of the active form of vitamin D in the liver. In this study, we found that *A. senticosus* could stimulate the active form of vitamin D in yellow catfish by increasing the transcript level of *CYP24a*. This led to reduced fat content in the liver. *PLPP3* encodes LPP3, a cell-surface enzyme and an intrinsic negative regulator that regulates the signals of biologically active lipids including lysophosphatidic acid [66]. Inhibition of *PLPP3* expression can lead to decreased lipid production in the body, thereby reducing the harmful effects of rapid fat deposition [67]. In this experiment, the transcript levels of *PLPP3* in the liver of yellow catfish were significantly reduced in the groups fed with diets containing 2 and 4 g/kg *A. senticosus*, and the fish in these groups also showed decreased HSI and lower TG and TC contents, compared with those in other groups and the control group. This result suggests that *A. senticosus* regulates the expression of *PLPP3* with an overall effect to reduce fatty deposits. Bae [68] detected significant expression of *PLPP3* in mice fed with high-fat diets. The gene *DIO2* encodes deiodinase type II (D2), which catalyzes the conversion of T4 to T3 [69]. In our study, we detected down-regulation of *DIO2* in the groups fed with diets containing 2 and 4 g/kg *A. senticosus*. This suggested that the pathway of T4 conversion to T3 was inhibited, consistent with the increased serum T4 content and decreased serum T3 content. In another study, up-regulation of *DIO2* was detected in the subcutaneous and visceral fat cells of obese patients [70]. Previous studies have shown that increased *DIO2* expression is related to decreased mitochondrial gene

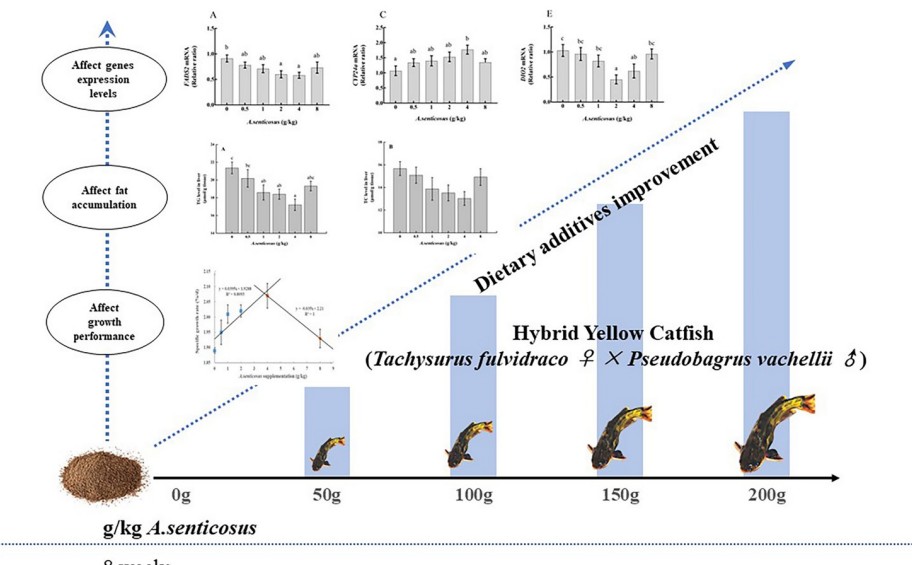

**Fig 7. Pathways potentially affected by *A. senticosus* supplementation in the diet of hybrid yellow catfish.** These pathways are related to growth performance and fat metabolism and involve genes that were differentially expressed in response to *A. senticosus* supplementation.

expression and lipid oxidation, and that mitochondrial dysfunction leads to decreased fatty acid oxidation [71] and increased lipid accumulation [72]. Our results indicate that *A. senticosus* can regulate the expression of *DIO2* to reduce fat accumulation in the liver tissue of yellow catfish.

## Conclusion

Our results show that appropriate dietary supplementation with *A. senticosus* can effectively promote the growth of hybrid yellow catfish, regulate its serum lipid levels, relieve excessive fat deposition, and protect the liver. The results of transcriptome analyses indicate that dietary *A. senticosus* affects the transcript levels of *FADS2*, *ELOVL2*, *CYP24a*, *PLPP3*, *DIO2*, thereby regulating lipid metabolism and reducing damage caused by fat deposition in the liver (Fig 7). These positive effects may be related to polysaccharides, flavonoids, and saponins in *A. senticosus*, but further research is required to clarify the specific roles of the active components of *A. senticosus* in fish. On the basis of our results, we suggest that the addition of 2–4 g/kg *A. senticosus* to the diet is beneficial for the health of farmed hybrid yellow catfish.

## Acknowledgments

We thank Jennifer Smith, PhD, from Liwen Bianji, Edanz Group China (www.liwenbianji.cn/ac), for editing the English text of a draft of this manuscript.

## Author Contributions

**Conceptualization:** Jun Qiang, Pao Xu.

**Data curation:** Ming Xiao Li, Yi Fan Tao, Hao Jun Zhu.

**Formal analysis:** Ming Xiao Li.

**Funding acquisition:** Pao Xu.

**Investigation:** Ming Xiao Li.

**Methodology:** Ming Xiao Li, Jing Wen Bao, Yi Fan Tao, Hao Jun Zhu.

**Project administration:** Ming Xiao Li, Jun Qiang, Jing Wen Bao.

**Resources:** Jun Qiang, Pao Xu.

**Supervision:** Jun Qiang, Jing Wen Bao, Yi Fan Tao, Hao Jun Zhu.

**Validation:** Ming Xiao Li.

**Writing – original draft:** Ming Xiao Li.

**Writing – review & editing:** Jun Qiang, Pao Xu.

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
