## [Decision Letter · Decision Letter 0]

16 Nov 2020

PONE-D-20-30481

Growth Performance, Physiological Parameters, and Transcript Levels of Lipid Metabolism-related Genes in Hybrid Yellow Catfish (Tachysurus fulvidraco ♀ × Pseudobagrus vachellii ♂) Fed with Diets Containing Acanthopanax senticosus

PLOS ONE

Dear Dr. Li,

Thank you for submitting your manuscript to PLOS ONE. After careful consideration, we feel that it has merit but does not fully meet PLOS ONE’s publication criteria as it currently stands. Therefore, we invite you to submit a revised version of the manuscript that addresses the points raised during the review process.

We look forward to receiving your revised manuscript.

Kind regards,

Silvia Martínez-Llorens

Academic Editor

PLOS ONE

Additional Editor Comments:

Following the suggestion raised by the reviewer, it could be also interesting to analyze the relation between digestive enzymes and protein efficiency ratio.

Journal Requirements:

2. In your Methods section, please provide additional information on the animal research and ensure you have included details on : (1) methods of sacrifice (2) methods of anesthesia and/or analgesia, and (2) efforts to alleviate suffering.

Reviewers' comments:

Reviewer's Responses to Questions

**Comments to the Author**

1. Is the manuscript technically sound, and do the data support the conclusions?

Reviewer #1: Yes

2. Has the statistical analysis been performed appropriately and rigorously? 

Reviewer #1: Yes

3. Have the authors made all data underlying the findings in their manuscript fully available?

Reviewer #1: Yes

4. Is the manuscript presented in an intelligible fashion and written in standard English?

Reviewer #1: Yes

5. Review Comments to the Author

Reviewer #1: Growth Performance, Physiological Parameters, and Transcript Levels of Lipid

Metabolism-related Genes in Hybrid Yellow Catfish (Tachysurus fulvidraco ♀ ×

Pseudobagrus vachellii ♂) Fed with Diets Containing Acanthopanax senticosus

The present study conducted to evaluate the effect Siberian ginseng in 5levelsongrowthandphysiological performance as well as metabolism related genes in hybrid yellow catfish. Dietary levels of 2-4 g/kg showed growth promotion and reduced cholesterol and triglycerides in serum and liver. Also dietary ginseng revealed down regulation of some related gene to reduce the fat deposition in the liver.

Is this fish a kind of low fat fish which can be expressed by aquaculture activities to be high fat content? Market always prefers original taste.

The author stated that amylase, trypsin, and lipase activities in the yellow catfish intestine significantly increased with increasing of A. senticosus, but no change in FCR was found. This should be discussed appropriately.

The authors said that they measured glycogen (page 6, line 139), but no data is available in this parameter?

The present of some antinutrients in Chinese ginseng could probably affect the results and this can be considered for more discussion.

The manuscript can be accepted, but there are some minor corrections to improving the manuscript before publication.

Some of my comments are listed below:

Abstract:

Give the duration of the experiment here.

Also some numerical data should be given in summary.

Keywords:

4 of 7 exactly repeated in the title and should be moderately changed.

Introduction:

Well writing. Just some explanation on the measured parameters in this study could mature the introduction.

The culture importance and problems in the examined fish should be given.

Materials and methods:

Page 4, line 90: use the same unit for g/kg throughout the text.

Page 4, line 95-97: the preparation of the diets is not complete in terms of the amount of added water and oil, time for drying and so on.

Page 4, line 108-109: give the other water physio-chemical parameters. The tank dimension, source of water and flow rate is not given!

Page 8, line 196: the company info for this version of SPSS in not correct.

Results:

Well writing.

Give the final weight of fish for better comparison by readers.

Discussion:

Page 13, line 292: change rate to level.

Page 13, line 296-298: but no changes were observed in FCR.

Page 13, line 299: should be present as strongest effect on…

Page 13, line 300-302: not related to the changes of enzymes. Some ingredients and the levels should be an effective reason.

Page 13, line 306-308: but the authors did not measure the catabolic enzymes.

Page 13, line 309-310: this sentence is repeated in introduction.

Page 14, line 336: change to HSI.

Page 15, line 369-370: I'm not agree with this as those are antinutrients and many studies showed the negative impacts of these on growth and metabolism of animals.

Reference:

There are many mistakes in references in terms of format, journal abbreviations, scientific names, etc. such as lines 394-395, 411,414-415,417, 421-422, and so on.

Tables:

Table 1: give the unit in parenthesis in front of the ginseng. Correct middling with small letter. Put all proximate analyses of the diets in terms of ash, dry matter and other items.

Table 3: insert the initial and final fish weights.

Table 4: why no data on T3, T3 and the ratio of T3:T4 are available in this table!? Those should be given here. The unit of TP is generally g/dL.

Figures:

The quality of all figures is low in the reviewers' file.

Figure 1: no data are available to how the broken-line was analyzed.

Figure 2: change value to ratio.

6. PLOS authors have the option to publish the peer review history of their article (what does this mean?). If published, this will include your full peer review and any attached files.

Reviewer #1: **Yes: **Bahram Falahatkar

---

## [Author Response · Author response to Decision Letter 0]

23 Dec 2020

We thank the reviewers for all their insightful comments, which have helped us greatly improve the manuscript. Our responses to all the reviewers’ comments are detailed below. We have reformatted the manuscript to meet PLOS ONE style requirements.

Comment 1: Following the suggestion raised by the reviewer, it could be also interesting to analyze the relation between digestive enzymes and protein efficiency ratio.

Response: In previous studies, we found that dietary A. senticosus supplementation increased the activity of intestinal digestive enzymes (amylase, trypsin, lipase) of hybrid yellow catfish. The organic substances (protein, fat, carbohydrate) in the diet are digested in the digestive tract, and then hydrolyzed into smaller molecules that are easily absorbed and used in the fish body. The digestive enzymes play important roles in this process. Previous studies have shown that, in fish, the digestion and absorption of dietary protein can be improved by increasing trypsin activity, thereby promoting the synthesis of fish protein and promoting its growth [1, 2].

We have added the protein efficiency ratio data (PER) in Table 3, and discussed the difference of PER at dietary A. senticosus supplementation in the results section (page 8, lines 154-155), and discussed the results in the discussion section (page 17, lines 369-373).

Comment 2: Is this fish a kind of low fat fish which can be expressed by aquaculture activities to be high fat content? Market always prefers original taste.

Response: Yellow catfish is an important freshwater economic fish in China. It has delicious meat and high nutritional value, and is loved by consumers. Because of the high demand for yellow catfish, the natural catch cannot meet the needs of consumers, so farmers are pursuing a higher-yield aquaculture model for profit. (According to statistics from the China Fisheries Yearbook, the total yield of yellow catfish in China in 2018 and 2019 was 509,610 and 536,964 tons, respectively, an increase of 5.37%[3].) Feeding large amounts of formulated feed can easily lead to excessive energy intake and fat deposition, and the excess fat may accumulates on muscle, but mainly in the liver and abdominal cavity. Fat deposition in the liver can increase the pressure on the liver, damage the cell structure, and affect liver metabolism, leading to slow growth and weakened immunity of farmed fish [4-6] and causing huge economic losses. 

Siberian ginseng (Acanthopanax senticosus) is rich in nutrients such as trace elements and amino acids, which are required by animals, as well as a variety of other active ingredients [7, 8], and A. senticosus also plays a role in regulating lipid metabolism [9, 10]. Therefore, to tackle some of the various problems caused by intensive aquaculture, we explored the mechanism by which dietary A. senticosus supplementation affects lipid metabolism in the liver of hybrid yellow catfish.

Comment 3: The authors said that they measured glycogen (page 6, line 139), but no data is available in this parameter?

Response: We have shown the glycogen data in Fig 3 and discussed liver glycogen content in the discussion section (page 18, lines 401-403).

Comment 4: The present of some antinutrients in Chinese ginseng could probably affect the results and this can be considered for more discussion.

Response: We have addressed this concern in our detailed response to comment 22 below.

Comment 5: Abstract- Give the duration of the experiment here.

Response: We have included this information in the revised abstract (line 16).

Comment 6: Abstract- Also some numerical data should be given in summary. 

Response: We have included some numerical data in the revised abstract (lines 17–21 and 24–27).

Comment 7: Keywords- 4 of 7 exactly repeated in the title and should be moderately changed.

Response: We have changed the keywords to avoid repetition (line 34–35).

Comment 8: Introduction- Just some explanation on the measured parameters in this study could mature the introduction.

Response: We have added explanations about the triglyceride (TG), total cholesterol (TC), triiodothyronine (T3), thyroxine (T4), and catabolic enzymes lipoprotein lipase (LPL) and hepatic lipase (HL) indicators in the introduction section (page 5, lines 88-105). Because we analyzed and screened the enrichment pathways and differentially expressed genes based on the results of transcriptome sequencing, and selected five differentially expressed genes related to lipid metabolism, we discussed these five genes in the discussion section.

Comment 9: Introduction- The culture importance and problems in the examined fish should be given.

Response: We mentioned current aquaculture problems in the introduction section (page 4, lines 46–51) and added relevant information about yellow catfish farming (page 4, lines 69–72).

Comment 10: Page 4, line 90- use the same unit for g/kg throughout the text.

Response: We have used the same unit throughout the revised manuscript (page 6, line 113).

Comment 11: Page 4, line 95-97- the preparation of the diets is not complete in terms of the amount of added water and oil, time for drying and so on.

Response: The content of each component in the feed is given in Table 1. We added relevant content information to the methods section under ‘Experimental diets’ (page 6, lines 118, 120–121).

Comment 12: Page 4, line 108-109- give the other water physio-chemical parameters. The tank dimension, source of water and flow rate is not given!

Response: We have added this information to the revised manuscript (page 7, lines 130, 133–135).

Comment 13: Page 8, line 196- the company info for this version of SPSS in not correct. 

Response: We have included this information as “SPSS ver. 22.0 (IBM Corp., Armonk, NY, USA)” (page 10, lines 223–224). 

Comment 14: Results- Give the final weight of fish for better comparison by readers.

Response: We have included the initial and final weights in Table 3. 

Comment 15: Page 13, line 292- change rate to level.

Response: We changed “rate” to “level” (page 16, line 361).

Comment 16: Page 13, line 296-298- but no changes were observed in FCR.

Response: We found that A. senticosus supplementation significantly enhanced intestinal digestive enzyme activities. Feed conversion rates (FCRs) were not significantly different, but they showed a downward trend with A. senticosus supplementation and the FCR was lowest with 4 g/kg A. senticosus. Jang et al. [11] found that dietary A. senticosus reduced the FCR of broiler chicks. Therefore, we considered that A. senticosus promoted the digestion and absorption of nutrients, but this result was not significant, likely because of differences between fish and chicken.

Comment 17: Page 13, line 299- should be present as strongest effect on…

Response: We changed the wording as suggested (page 17, line 368).

Comment 18: Page 13, line 300-302- not related to the changes of enzymes. Some ingredients and the levels should be an effective reason.

Response: The original statement was incorrect and has been deleted. A. senticosus contains a variety of nutrients and active ingredients, and has been shown to promote the development of animal intestines and promote secretion from digestive glands [12].

Comment 19: Page 13, line 306-308- but the authors did not measure the catabolic enzymes.

Response: To supplement the data about the catabolic enzymes, we measured the activities of lipoprotein lipase (LPL) and hepatic lipase (HL) in the liver homogenate using an ELISA kit according to the manufacturer’s instructions (Table 5). We also added information about LPL and HL in the introduction section (page 5, lines 94–98), included the enzyme activity test in the result section (page 13, lines 271–276), and discussed the results in the discussion section (page 17, lines 386–394).

Comment 20: Page 13, line 309-310- this sentence is repeated in introduction.

Response: We deleted the repeated sentence.

Comment 21: Page 14, line 336- change to HSI.

Response: This was changed (page 18, line 415).

Comment 22: Page 15, line 369-370- I'm not agree with this as those are antinutrients and many studies showed the negative impacts of these on growth and metabolism of animals.

Response: The A. senticosus samples were tested by the ZKGX Research Institute of Chemical Technology (Chemical Lab, Beijing, China) and were found to contain active ingredients such as polysaccharides, flavonoids, and saponins, whereas the content of lignans and other ingredients was low or undetectable. These active ingredients have been found to positively affect animal growth [13], enhance immunity [14], improve antioxidant capacity [15], treat inflammation [16, 17], and regulate lipid metabolism [9, 10]. 

Sui et al. [18] injected A. senticosus saponins into experimental hyperlipidemia rats and found that after 21 days, serum TG and TC levels decreased significantly and liver fat deposition was significantly reduced. Related studies have shown that plant active polysaccharides have a lipid-lowering function; for example, Astragalus membranaceus polysaccharides [19, 20] and Lycium barbarum polysaccharides [21] all show similar effects. Meng et al. [22] showed that A. senticosus polysaccharides have a positive effect on the inhibition of lipid peroxidation of the erythrocyte membrane. Wen et al. [23] found that the total flavonoids of A. senticosus increased erythrocyte membrane fluidity, reduced blood viscosity, and improved hemorheology factors.

On the basis of these results, we considered that the positive effects of A. senticosus supplementation on lipid-lowering and growth may be related to the polysaccharide, flavonoid, and saponin contents of A. senticosus.

Comment 23: Reference- There are many mistakes in references in terms of format, journal abbreviations, scientific names, etc. such as lines 394-395, 411,414-415,417, 421-422, and so on.

Response: We have reformatted the references to remove errors.

Comment 24: Table 1- give the unit in parenthesis in front of the ginseng. Correct middling with small letter. Put all proximate analyses of the diets in terms of ash, dry matter and other items.

Response: We modified Table 1 to include all the suggested changes.

Comment 25: Table 3- insert the initial and final fish weights.

Response: The initial and final fish weights have been included in Table 3.

Comment 26: Table 4- why no data on T3, T3 and the ratio of T3:T4 are available in this table!? Those should be given here. The unit of TP is generally g/dL.

Response: We have added the T3, T4, and T3/T4 data to Table 4, and plotted their levels in Fig 2. Serum total protein (TP) content was measured using a fully automatic biochemical analyzer (bs-400, MINDRAY, Shenzhen, China) according to the manufacturer’s instructions; the TP unit is given as g/L, so we did not change the unit. Qiang et al. [24, 25] and Bao et al. [26] also reported TP content in g/L.

Comment 27: Figures- The quality of all figures is low in the reviewers' file.

Response: We changed the figure mode to sRGB and increased the resolution to 600 dpi to improve the quality of the figures.

Comment 28: Figure 1: no data are available to how the broken-line was analyzed.

Response: We calculated the specific growth rate (SGR) of the hybrid yellow catfish and included the data in Table 3. We used a two-slope broken-line model to analyze the relationship between SGR and dietary supplementation with A. senticosus according to the method of Qiang et al. [27]. This information has been added to the legend for Fig 2.

Comment 29: Figure 2: change value to ratio.

Response: We changed “value” to “ratio” in Fig 2.

 

References:

1. Sun HC, Xu JM, Pang M. Effects of dietary protein levels on digestive enzyme activities of yellow catfish pelteobagrus vachelli. J. Hydroecol. 2010; 3(2): 84-88. https://doi.org/10.15928/j.1674-3075.2010.02.016.

2. Huang WQ, Sun YP, Wang GX, Wang ST, Mo WY, Chen XY, et al. Effects of ethanol extract from Chlorella vulgaris on growth performance, body composition, digestive enzyme activities and serum biochemical and antioxidant indices of juvenile pelteobagrus vachelli. Chin. J. Anim. Nutr. 2020; 32(4): 1778-1787. https://doi.org/10.3969/j.issn.1006-267x.2020.04.036.

3. China Fishery Statistical Yearbook. China Fishery Statistical Yearbook. Beijing: China Agriculture Press. 2020.

4. Du ZY, Clouet P, Zheng WH, Degrace P, Tian LX, Liu YJ. Biochemical hepatic alterations and body lipid composition in the herbivorous grass carp (Ctenopharyngodon idella) fed high-fat diets. Br. J. Nutr. 2006; 95(5): 905-915. https://doi.org/10.1079/bjn20061733.

5. Lu KL, Xu WN, Li JY, Li XF, Huang GQ, Liu WB. Alterations of liver histology and blood biochemistry in blunt snout bream Megalobrama amblycephala fed high-fat diets. Fish. Sci. 2013; 79(4): 661-671. https://doi.org/10.1007/s12562-013-0635-4.

6. Li D, Liu LT. Introduction to diagnosis and treatment of fish fatty liver disease. Prog. Vet. Med. 2016; 37(1): 114-117. https://doi.org/10.3969/j.issn.1007-5038.2016.01.026.

7. Yi JM, Kim MS, Seo SW, Lee KN, Yook CS, Kim HM. Acanthopanax senticosus root inhibits mast cell-dependent anaphylaxis. Clin. Chim. Acta. 2001; 312(1): 163-168. https://doi.org/10.1016/S0009-8981(01)00613-1.

8. Yi JM, Hong SH, Kim JH, Kim HK, Song HJ, Kim HM. Effect of Acanthopanax senticosus stem on mast cell-dependent anaphylaxis. J. Ethnopharmacol. 2002; 79(3): 347-352. https://doi.org/10.1016/s0378-8741(01)00403-2.

9. Park SH, Lee SG, Kang SK, Chung SH. Acanthopanax senticosus reverses fatty liver disease and hyperglycemia in ob/ob mice. Arch. Pharmacal Res. 2006; 29(9): 768-776. https://doi.org/10.1007/bf02974078.

10. Li MX, Li HX, Qiang J, Xu P, Bao JW, Chen DJ, et al. Effects of Acanthopanax Senticosus on Growth Performance, Fat Deposition and Proinflammatory Cytokine Expression of Genetically Improved Farmed Tilapia (GIFT, Oreochromis Niloticus). Anim. Nutr. 2019; 31(12): 5801-5812. https://doi.org/10.3969/j.issn.1006-267x.2019.12.046.

11. Jang I, Moon YS, Hwan SS. Effect of Supplementation of Acanthopanax senticosus on Growth Performance,Blood Biochemical Profiles and Expression of Pro-Inflammatory Cytokinesin Broiler Chicks. Korean J. Poult. Sci. 2015; 42(3): 197-204. https://doi.org/10.5536/KJPS.2015.42.3.197.

12. Li Q, Ge C, Tian Y. Effect of Chinese Herb Feed Additives on the Growing Charateristics in Piglets(Ⅱ). J. Yunnan Agric. Univ. 2002; 17(1): 63-66. https://doi.org/CNKI:SUN:YNDX.0.2002-01-013.

13. Fang J, Yan FY, Kong XF, Ruan Z, Liu ZQ, Huang RL, et al. Dietary supplementation with Acanthopanax senticosus extract enhances gut health in weanling piglets. Livest. Sci. 2009; 123(2-3): 268-275. https://doi.org/10.1016/j.livsci.2008.11.020.

14. Chen RZ, Liu ZQ, Zhao JM, Chen RP, Meng FL, Zhang M, et al. Antioxidant and immunobiological activity of water-soluble polysaccharide fractions purified from Acanthopanax senticosu. Food Chem. 2011; 127(2): 434-440. https://doi.org/10.1016/j.foodchem.2010.12.143.

15. Lee SY, Son D, Ryu J, Lee YS, Jung SH, Kang JI, et al. Anti-oxidant activities of Acanthopanax senticosus stems and their lignan components. Arch. Pharmacal Res. 2004; 27(1): 106-110. https://doi.org/10.1007/bf02980055.

16. Yamazaki T, Shimosaka S, Sasaki H, Matsumura T, Tukiyama T, Tokiwa T. (+)-Syringaresinol-di-O-beta-D-glucoside, a phenolic compound from Acanthopanax senticosus Harms, suppresses proinflammatory mediators in SW982 human synovial sarcoma cells by inhibiting activating protein-1 and/or nuclear factor-kappa B activities. Toxicol. In Vitro. 2007; 21(8): 1530-1537. https://doi.org/10.1016/j.tiv.2007.04.016.

17. Jung J, Park HJ, Kim RG, Shin KM, Ha J, Choi JW, et al. In vivo anti-inflammatory and antinociceptive effects of liriodendrin isolated from the stem bark of Acanthopanax senticosus. Planta Med. 2003; 69(7): 610-616. https://doi.org/10.1055/s-2003-41127.

18. Sui DY, Han CC, Yu XF, Qu SC. Effects of Acanthopanax senticosus saponins on blood lipid metabolism and antioxidative activity in experimental hyperlipidemia rats. J. Jilin Univ (Med. Ed). 2004; 30(1): 56-59. https://doi.org/10.13481/j.1671-587x.2004.01.026.

19. Wu K, Ouyang J, Wu Y, Liu M, Mao XQ, Wang BH, et al. Insulin sensitization and anti-obesity effects of Astragalus polysaccharide on high fat diet-fed C57BL/6J mice. FASEB J. 2006; 20(5): A1146. https://doi.org/10.1111/j.1748-1716.1951.tb00787.x.

20. Huang YC, Tsay HJ, Lu MK, Lin CH, Yeh CW, Liu HK, et al. Astragalus membranaceus-Polysaccharides Ameliorates Obesity, Hepatic Steatosis, Neuroinflammation and Cognition Impairment without Affecting Amyloid Deposition in Metabolically Stressed APPswe/PS1dE9 Mice. International Int. J. Mol. Sci. 2017; 18(12): 1-17. https://doi.org/10.3390/ijms18122746.

21. Wu HT, He XJ, Hong YK, Ma T, Xu YP, Li HH. Chemical characterization of lycium barbarum polysaccharides and its inhibition against liver oxidative injury of high-fat mice. Int. J. Biol. Macromol. 2010; 46(5): 540-543. https://doi.org/10.1016/j.ijbiomac.2010.02.010.

22. Meng QF, Yu XK, Xu MY, Li ML, Gao ZH, Fan H, et al. Extraction of Acanthopanacis Senticosi Polysaccharides and their Antioxidative Effect. J. Jilin Univ (Med. Ed). 2005; 43(5): 683-686. https://doi.org/10.13413/j.cnki.jdxblxb.2005.05.031.

23. Wen XX, Liu W, Deng XJ, Zhao GR, Feng DH, Sun L. Determination of total flavonoids of Acanthopanax Senticosus extract and its effect on hemorheology. Pharm. J. Chin. PLA. 2006; 22(3): 197-199. https://doi.org/10.3969/j.issn.1008-9926.2006.03.012.

24. Qiang J, Yang H, Wang H, Kpundeh MD, He J, Xu P. Physiological responses and HSP70 mRNA expression in GIFT tilapia juveniles, Oreochromis niloticus under short-term crowding. Aquacult. Res. 2015; 46(2): 335-345. https://doi.org/10.1111/are.12189.

25. Qiang J, Wasipe A, He J, Tao YF, Xu P, Bao JW, et al. Dietary vitamin E deficiency inhibits fat metabolism, antioxidant capacity, and immune regulation of inflammatory response in genetically improved farmed tilapia (GIFT, Oreochromis niloticus) fingerlings following Streptococcus iniae infection. Fish Shellfish Immunol. 2019; 92: 395-404. https://doi.org/10.1016/j.fsi.2019.06.026.

26. Bao JW, Qiang J, Tao YF, Li HX, He J, Xu P, et al. Responses of blood biochemistry, fatty acid composition and expression of microRNAs to heat stress in genetically improved farmed tilapia (Oreochromis niloticus). J. Therm. Biol. 2018; 73: 91-97. https://doi.org/10.1016/j.jtherbio.2018.02.007.

27. Qiang J, Khamis OAM, Jiang HJ, Cao ZM, He J, Tao YF, et al. Effects of dietary supplementation with apple peel powder on the growth, blood and liver parameters, and transcriptome of genetically improved farmed tilapia (GIFT, Oreochromis niloticus). PloS One. 2019; 14(11): 1-22. https://doi.org/10.1371/journal.pone.0224995.

---

## [Editor Report · Decision Letter 1]

20 Jan 2021

Growth performance, physiological parameters, and transcript levels of lipid metabolism-related genes in hybrid yellow catfish (Tachysurus fulvidraco♀ × Pseudobagrus vachellii ♂) fed diets containing Siberian ginseng

PONE-D-20-30481R1

Dear Dr. Li,

We’re pleased to inform you that your manuscript has been judged scientifically suitable for publication and will be formally accepted for publication once it meets all outstanding technical requirements.

Kind regards,

Silvia Martínez-Llorens

Academic Editor

PLOS ONE
---

## [Editor Report · Acceptance letter]

2 Feb 2021

PONE-D-20-30481R1 

Growth performance, physiological parameters, and transcript levels of lipid metabolism-related genes in hybrid yellow catfish (*Tachysurus fulvidraco* ♀ × *Pseudobagrus vachellii* ♂) fed diets containing Siberian ginseng 

Dear Dr. Li:

I'm pleased to inform you that your manuscript has been deemed suitable for publication in PLOS ONE. Congratulations! Your manuscript is now with our production department. 

Kind regards, 

on behalf of

Dr Silvia Martínez-Llorens 

Academic Editor

PLOS ONE